# Exploring the Potential of Large Language Models in Generating Code-Tracing Questions for Introductory Programming Courses

**Aysa Xuemo Fan[1], Ranran Haoran Zhang[2], Luc Paquette[1], Rui Zhang[2]**
[1] University of Illinois at Urbana-Champaign
[2] Penn State University
{xuemof2, lpaq}@illinois.edu
{hzz5361, rmz5227}@psu.edu

## Abstract

In this paper, we explore the application of large language models (LLMs) for generating code-tracing questions in introductory programming courses. We designed targeted prompts for GPT4, guiding it to generate code-tracing questions based on code snippets and descriptions. We established a set of human evaluation metrics to assess the quality of questions produced by the model compared to those created by human experts. Our analysis provides insights into the capabilities and potential of LLMs in generating diverse code-tracing questions. Additionally, we present a unique dataset of human and LLM-generated tracing questions, serving as a valuable resource for both the education and NLP research communities. This work contributes to the ongoing dialogue on the potential uses of LLMs in educational settings[1].

## 1 Introduction and Background

The teaching of introductory programming courses continues to be a challenging endeavor, despite the global uptake and popularity of such courses. High enrollment rates often result in diverse student populations, with a wide range of programming experience from those just starting their journey to others with prior exposure (Lopez et al., 2008). Ensuring an effective learning experience that accommodates this wide disparity presents a daunting task, making the teaching of these courses complex.

One critical component in teaching introductory programming is the focus on code tracing, a skill identified as instrumental in enhancing code writing abilities (Lister et al., 2009; Venables et al., 2009; Kumar, 2013). Current educational methodologies encourage code tracing through a variety of means, such as practice questionnaires (Lehtinen et al., 2023), direct teaching strategies (Xie et al.,

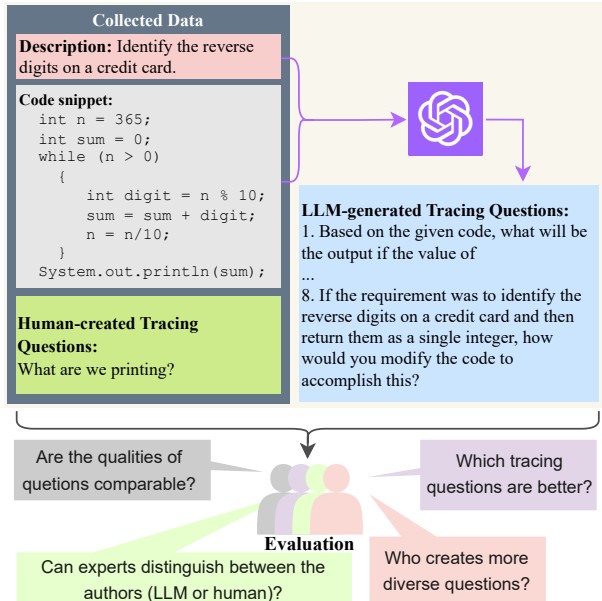

Figure 1: We aim to assess the Large Language Models' (LLMs') capability to generate code tracing questions, pivotal in computer science education. The accompanying illustration outlines our approach, where GPT4 is employed to generate questions based on given code snippets and descriptions. Subsequent comparative analysis with human-created questions aids in exploring critical aspects, such as the quality and diversity of generated questions, discernibility between human and AI authors, and the relative superiority in question quality.

2018), and tracing quizzes (Sekiya and Yamaguchi, 2013). These strategies consistently utilize code-tracing questions aimed at fostering and developing a student's understanding and skills.

However, the preparation of code-tracing questions poses challenges. Manual question creation by instructors (Sekiya and Yamaguchi, 2013; Hassan and Zilles, 2021) is time-consuming and lacks scalability. Automatic generation using program analysis saves time, yet is limited by the analyzer's capabilities and lacks question diversity (Zavala and Mendoza, 2018; Thomas et al., 2019; Russell, 2021; Lehtinen et al., 2021; Stankov et al., 2023).

---

[1] Our data and code are available at https://github.com/aysafanxm/llm_code_tracing_question_generation

In light of the increasing potential of Large Language Models (LLMs) in sectors like code summarization and explanation (Chen et al., 2021; Siddiq et al., 2023), the question arises: Can LLMs generate high-quality code-tracing questions? Our study explores this query using GPT4 (OpenAI, 2023), leveraging prompts to guide its question generation based on given code snippets and descriptions. To assess the LLM's capability in this pivotal aspect of computer science education, we devised a set of human evaluation metrics. This allowed for an objective appraisal of the LLM-generated questions, and, through a comparative analysis with human-created counterparts, critical aspects such as question quality, diversity, discernibility between human and AI authors, and relative superiority in quality were explored (Figure 1). These analyses have enhanced our understanding of the potential roles of LLMs in computer science education.

This investigation provides a foundation for considering the potential inclusion of LLMs in learning platforms, which could offer new possibilities for enhancing the learning experience in introductory programming courses. Given these advancements, our study contributes to the field as follows:

- The curation of a high-quality dataset consisting of human and LLM-generated code tracing questions and associated code snippets.

- An exploration and evaluation of GPT4's capability in question generation, including comparisons with both GPT3.5-turbo and human-authored questions, and an examination of few-shot and zero-shot scenarios.

- The introduction of a human evaluation methodology and a comprehensive assessment of the quality of LLM-generated questions, offering valuable insights into the potential of LLMs in educational contexts.

## 2 Related Work

**Question Generation:** Early Question Generation (QG) research primarily focused on multiple-choice questions (Mitkov et al., 2006; Agarwal and Mannem, 2011) and questions with specific interrogatives (Heilman and Smith, 2010). With the emergence of the SQuAD dataset (Rajpurkar et al., 2016), context-dependent QG gained momentum (Du et al., 2017; Yuan et al., 2017; Subramanian et al., 2018; Puri et al., 2020). This extended to

complex tasks like generating unanswerable questions (Choi et al., 2018; Zhu et al., 2019; Reddy et al., 2019) and multi-hop reasoning (Pan et al., 2020, 2021; Shridhar et al., 2022). Our work, focusing on generating code tracing questions in CS Education domain, addresses unique challenges around code, natural language, and pedagogical comprehension, inadequately covered by previous methods due to a lack of specialized datasets.

**Code LLMs for CS Education:** Recent advances in code large language models (LLMs) (Chen et al., 2021; Wang et al., 2021; Le et al., 2022; Wang et al., 2023) have enabled various downstream applications, including code completion, retrieval, summarization, explanation, and unit test generation (Lu et al.; Siddiq et al., 2023; Tian et al., 2023). Studies have showcased the LLMs' ability to generate novice programming content comparable to humans (Finnie-Ansley et al., 2022; Piccolo et al., 2023). LLMs have been utilized in classroom environments (Kazemitabaar et al., 2023), to generate coding exercises and explanations (Sarsa et al., 2022), and to create counterfactual questions (Narayanan et al., 2023). Our study represents the first exploration of LLMs for code tracing question generation, a critical component of CS Education, thus underscoring the potential of these models for generating educational content.

## 3 Our Approach

### 3.1 Task Definition

In automatic tracing question generation, given a description (optional) $d \in D \cup \emptyset$, detailing the code context, and a code snippet $c \in C$ provided by an instructor or student, the aim is to generate a set of relevant questions $Q'$ for student practice. This task can be formally defined as a function:

$$f : (d, c) \mapsto Q' \qquad (1)$$

where $D$ represents all possible descriptions, $C$ all possible code snippets, and $Q'$ is a subset of all possible questions $Q$.

### 3.2 Curating the Code-Tracing Question Dataset

For our experiment, we curated a unique dataset reflecting the range of tracing questions encountered by beginner programmers. We sourced 158 unique

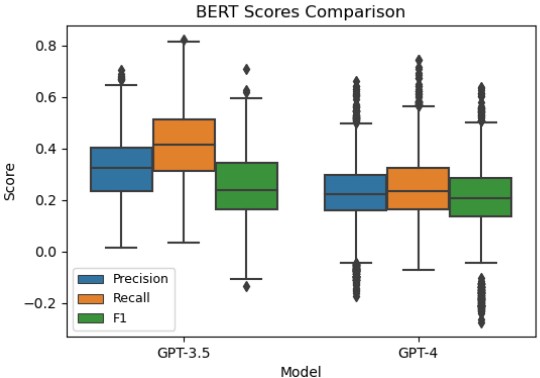

Figure 2: Comparison of the BERTScore on all LLM-generated questions and human-authored questions.

questions from CSAwesome[2], a recognized online Java course aligned with the AP Computer Science A curriculum. To enhance diversity, we added 18 questions extracted from relevant YouTube videos. Other platforms and sources were also examined but didn't fit due to a lack of explicit tracing questions. Our final dataset consists of 176 unique code snippets and question pairs, allowing a fair evaluation of LLMs' ability to generate introductory programming tracing questions.

### 3.3 Prompt Engineering and Model Selection

In our iterative approach to prompt engineering and model selection, we first refined prompts and then generated tracing questions using GPT-3.5-turbo and GPT-4. Using BERTScore, we assessed question diversity and similarity. Based on these insights, we combined the optimized prompt with the chosen model to determine the most effective generation approach, be it few-shot or zero-shot. Our final prompt, refined iteratively from (Brown et al., 2020), positioned in Appendix B, adopts an expert instructor's perspective, encourages deep understanding via code-tracing questions, and maximizes the inherent versatility of LLMs.

Next, we considered GPT-3.5-turbo and GPT-4 for model selection, and investigate the generated tracing questions diversity by BERTScore (Zhang* et al., 2020). Regarding the automatic evaluation of the diversity in generated questions, we adopted the following methodology: For each code snippet, we utilize a singular human-authored tracing question as the reference. Both GPT3.5-turbo and GPT4 are then tasked with generating multiple tracing questions for every snippet. Following this,

we employ regular expressions in a postprocessing step to segment the generated content, isolating individual tracing questions. Subsequently, for each generated prediction $p$, its BERTScore is computed in relation to the reference, denoted as BERTScore(reference, $p$).

The boxplot in Figure 2 displays the Precision, Recall, and F1 scores for both models. From the graph, it's clear that GPT-3.5-turbo has a median Precision score around 0.45, Recall slightly above 0.6, and an F1 score hovering around 0.5. In comparison, GPT-4 shows a more balanced performance with a median Precision score close to 0.6, Recall near 0.55, and F1 just above 0.5. Notably, the variability in scores, particularly for GPT-4, highlights the diverse outcomes in its results. Based on our results, we chose GPT4 for subsequent evaluations. Enhanced performance examples from GPT4 are in Appendix C.

Next, we hypothesized that the few-shot question generation approach, which feeds the model with three tracing question examples and their respective code snippets, would yield higher-quality questions than the zero-shot generation that relies solely on the prompt. Contrary to our expectations, the experiment showed that the few-shot method introduced a significant bias towards the example questions, thus narrowing the diversity in the generated questions. Consequently, we opted for the zero-shot generation in our tests, which fostered a broader spectrum of question types. Detailed examples of outcomes from both the zero-shot and few-shot approaches are available in Section 4.4.

### 3.4 Human Evaluation

Next, we conducted a human evaluation comparing the quality of GPT4-generated and human-authored tracing questions. The expert evaluators were meticulously screened based on specific criteria: they had to be computer science graduate students with at least one year of programming teaching or tutoring experience. Four such experts, meeting these criteria, participated in the evaluation.

Each evaluator was assigned a set of 44 randomly selected code snippets from a pool of 176 human-authored tracing questions. For each snippet, evaluators received a pair of questions (one human-authored and one GPT4-generated) in a randomized order to mitigate potential ordering bias. Evaluators unawareness of question authorship was

---

[2]https://runestone.academy/ns/books/published/csawesome/index.html

| Criteria | Label |
|---|---|
| Relevance to Learning Objectives | 1-5 |
| Tracing or not | Yes/No |
| Clarity of the Question | 1-5 |
| Difficulty Level | 1-5 |
| Relevance to the Given Code Snippet | 1-5 |
| Ability to Distinguish Source | Human-created/ AI-generated |
| Preference for Better Question | Check preferred |

Table 1: Criteria used for expert evaluation.

| | Human | | GPT4 | |
|---|---|---|---|---|
| | Mean | Median | Mean | Median |
| Relevance to Learning | 4.78 | 5.00 | 4.62 | 5.00 |
| Question Clarity | 4.72 | 5.00 | 4.42 | 5.00 |
| Appropriate Difficulty | 4.75 | 5.00 | 4.43 | 5.00 |
| Relevance to Code | 4.72 | 5.00 | 4.64 | 5.00 |

Table 2: Comparative statistics for human and GPT4 generated questions.

ensured.

The evaluators rated each question based on five criteria shown in Table 1. They also guessed the question's authorship and expressed their preference between the pair. Detailed evaluation criteria and labels can be found in Table 1.

## 4 Analyses and Results

This section details our analysis and highlights the results, encompassing quality ratings, expert perceptions, and textual similarities in question generation.

### 4.1 Comparative Analysis of Quality Ratings

| | U-val | $p$ |
|---|---|---|
| Relevance to learning objectives | 3688.0 | 0.047 |
| Question Clarity | 3392.0 | 0.011 |
| Difficulty Appropriateness | 3540.5 | 0.015 |
| Relevance to the given code snippet | 3918.5 | 0.595 |

Table 3: Results of Mann-Whitney U tests comparing human evaluations of LLM-generated and human-authored code-tracing questions. U-values and p-values are provided for four criteria. Significant differences for three criteria indicate varying performance, while no difference in 'relevance to the given code snippet' suggests similar performance.

To assess the quality disparity between LLM-generated and human-authored questions, we applied Mann-Whitney U tests (Mann and Whitney, 1947) to the median ratings of four evaluation criteria in Table 3. Significant differences emerged in three criteria: *relevance to learning objectives*, *clarity*, and *difficulty appropriateness*. However, the *relevance to the given code snippet* showed no significant difference, indicating comparable performances.

Despite U-tests highlighting significant differences in some criteria, the practical quality difference was minimal. As further detailed in Table 2, LLM-generated questions had slightly lower mean ratings, yet their median ratings closely mirrored those of human-authored questions.

Considering these two analyses together, it is apparent that despite some statistical differences, LLM-generated questions still maintain a high pedagogical standard effectively. Consequently, LLM, while underlining areas for potential enhancement, demonstrates proficiency in generating questions that align closely in quality and course relevance with those crafted by humans.

### 4.2 Expert Perception of Question Authorship

We further evaluated the discernibility of LLM-generated questions from human-authored ones using a Confusion Matrix (Table 5). Approximately 56% (99 out of 176) of GPT4-generated questions were mistakenly identified by experts as human-generated, and about 20% (35 out of 176) of human-authored questions were misattributed to the GPT4. This overlap signifies the high quality of the generated questions and GPT4's proficiency in producing pedagogically relevant tracing questions. Moreover, the matrix reveals an evaluator bias toward attributing higher-quality questions to human authorship.

### 4.3 Textual Similarity between Questions

Table 6 presents BLEU (Post, 2018), ROUGE-1/2/L (Lin, 2004), and BERTScores (Zhang* et al., 2020), comparing the similarity between the randomly selected GPT4 generated questions and corresponding human-authored questions. The low BLEU and ROUGE scores suggest that GPT4 is generating distinct, non-verbatim questions compared to human-authored questions. A moderate BERTScore, reflecting semantic similarity, suggests that GPT4-generated questions align with the context of human-authored ones. This further underscores GPT4's capability to independently generate relevant and diverse code-tracing questions, distinct from those crafted by humans.

| GPT-4: Few-Shot vs. Zero-Shot Results | |
|---|---|
| **Zero-Shot Prompt:** | **Few-Shot Prompt:** |
| Your task ... Craft tracing questions for: {requirement} {code_snippet} | Your task ... Examples: {requirement1} {code_snippet1} {tracing_question1} ... Craft for: {requirement} {code_snippet} |
| **Zero-Shot Result:**
1. What is the purpose of this function? What does it aim to accomplish?
2. What is the base case for this recursive function? What happens when the base case is reached?
3. If the input values are i = 8 and j = 2, what will be the output of the function? Please trace through the code step-by-step and explain your reasoning.
...
6. What happens if both input values are negative, such as i = -8 and j = -2? Does the function handle this case correctly according to the requirement? Explain your reasoning. | **Few-Shot Result:**
Tracing question 1: What is the output of the method redo(9, 3)?
Tracing question 2: What is the output of the method redo(0, 5)?
Tracing question 3: What is the output of the method redo(27, 3)?

... |

Table 4: Illustrative comparison between GPT-4's code-tracing question generation in Few-Shot and Zero-Shot settings, showcasing the diversity and specificity of generated questions.

|  | Predicted GPT4 | Predicted Human |
|---|---|---|
| Actual GPT4 | 77 | 99 |
| Actual Human | 35 | 141 |

Table 5: Confusion Matrix indicating experts' attributions of code-tracing questions. The table displays the number of actual GPT-4 generated and human-authored questions and how they were predicted by the experts, underscoring the challenge in distinguishing between the two.

Combining with the previous analyses, the LLM, such as GPT4, thus exhibits substantial promise in generating high-quality, course-relevant code-tracing questions, illustrating its utility as a teaching aid.

## 4.4 Few-shot vs Zero-shot Generation Results

Few-shot generation biased our model towards the provided examples, largely reducing question diversity. In contrast, zero-shot generation yielded more diverse questions, prompting us to favor it for broader question variety in our experiment. Detailed examples of the generated results for both 0-shot and few-shot methods can be found in Appendix D.

Table 4 provides a side-by-side comparison of GPT-4's performance in few-shot and zero-shot settings. The zero-shot results exhibit a broader range of question types, while the few-shot results seem to be more templated, reflecting the bias introduced by the provided examples.

Possible reasons for these observations include

| Metric | Score |
|---|---|
| BLEU | 0.02 |
| ROUGE-1 F-score | 0.215 |
| ROUGE-2 F-score | 0.051 |
| ROUGE-L F-score | 0.199 |
| BERTScore Precision | 0.274 |
| BERTScore Recall | 0.341 |
| BERTScore F1 | 0.303 |

Table 6: Comparing the similarity between human-authored questions and the random choiced GPT4 question.

the influence of training data and model design in zero-shot scenarios, allowing GPT-4 to tap into its vast training experience. In contrast, in few-shot scenarios, the model might overly adhere to the provided examples, interpreting them as stringent templates, which can compromise output diversity. The balance between the nature of the task and the examples becomes pivotal in few-shot settings, potentially leading to outputs that may sacrifice accuracy or diversity. These hypotheses warrant further investigation in future work.

## 5 Conclusion

This study explored the capability of GPT-4 in generating code-tracing questions that rival the quality of those crafted by human educators. The findings illuminate the potential of LLMs to bolster programming education, marking a significant stride in the domain of code-tracing question generation and LLM application. This sheds light on scalable, high-quality automated question generation.

## Limitations and Future Work

This study marks a step closer in evaluating LLMs for code tracing question generation, but it is not without its limitations. Our research was primarily anchored to GPT-4, raising concerns about the generalizability of our findings to other LLMs, such as CodeT5+. Moreover, the study did not delve into the personalization of tracing questions based on individual student submissions, a facet that could greatly enhance the learning experience. Furthermore, the real-world educational efficacy of the LLM-generated questions remains an open question, given that our study did not involve actual students.

Several avenues beckon for further exploration. Evaluations with a broader range of models will offer a more comprehensive perspective on LLM capabilities. While our study centered on introductory Java tracing questions, assessing LLM versatility across different programming domains is imperative. The potential of LLMs extends beyond mere question generation; by tailoring questions to student needs, we can amplify the educational relevance. Our roadmap includes the development of an educational platform integrated with LLM questions, followed by classroom experiments and usability testing. To ensure broader applicability, expanding our dataset is crucial. Lastly, our findings on few-shot and zero-shot learning necessitate further investigation into model adaptability, biases in question generation, and the potential of intermediate-shot learning.

These directions not only underscore the transformative potential of LLMs in AI-driven education but also emphasize the importance of comprehensive evaluations.

## Ethical Statement

Our exploration of Large Language Models (LLMs) in introductory programming education was conducted ethically. We sourced public data and maintained evaluator anonymity and data confidentiality through secure storage. Evaluators were informed of the objectives and participated voluntarily. All evaluation results, as committed in the IRB forms, are securely stored. We strived for educational fairness by properly compensating the educators involved in our evaluation. We are mindful of the societal impacts of LLM integration in education. While acknowledging their promise, we believe careful consideration of pedagogical goals within the educational ecosystem is vital. Our future work will be guided by these ethical principles of privacy, informed consent, secure data handling, inclusivity, and conscientious progress focused on students' best interests.

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

## A Questionare

---

**Description**: (blank)
**Code**:

```java
int[][] m = {{1,1,1,1},{1,2,3,4},{2,2,2,2},{2,4,6,8}};
int sum = 0;
for (int k = 0; k < m.length; k++) {
    sum = sum + m[m.length-1-k][1];
}
```

**Question 1**: Given the following code segment, what is the value of sum after this code executes?
**Question 2**: What is the role of the expression 'm[m.length-1-k][1]' in the code?
**Annotation**
1. Relevance to Learning Objectives: The question is relevant to the learning objectives of an introductory programming course. (Label: 1-5)
2. Tracing or not: Is this a tracing question? (Label: Yes or No)
3. Clarity of the Question: The question presented is clear and the language used in the question is easy to understand. (Label: 1-5)
4. Difficulty Level: The difficulty level of the question is appropriate for an introductory programming course. (Label: 1-5)
5. Relevance to the Given Code Snippet: The question is appropriately related to the code snippet provided in the question. (Label: 1-5)
6. Ability to Distinguish Human-Authored from Automatically Generated Questions: Can you tell if the question is human-authored or automatically generated? (Label: Human-created or AI-generated)
7. I think this is a better tracing question. (Check the box under the better question)

---

This is an example of our questionnaire sent to annotators.

## B Prompts

The final prompt we employed offers LLMs a detailed context: it requests the generation of questions from an expert instructor's perspective within a defined pedagogical setting. It outlines the merits of code-tracing questions, gives an insight into their typical structure, and highlights their educational aim, specifically, promoting in-depth understanding rather than just assessing knowledge. Unlike the data collection process, where each code snippet is linked to a single question, the prompt is designed to produce multiple valid tracing questions for the same snippet, which leverages the inherent diversity and breadth potential of LLMs.

**User Prompt:** In your role as an education expert in an introductory Java programming course, you are equipped with a deep understanding of Java and teaching methodologies. Your aim is to shape the minds of young learners, paving their path to becoming proficient programmers. One of your potent teaching techniques involves providing students with real-world coding practice requirements and letting them craft working code snippets.

Consider a scenario where you have presented your students with a requirement to develop a basic calculator program in Java, performing fundamental arithmetic operations like addition, subtraction, multiplication, and division. Enthusiastically, the students pour their effort into crafting varied solutions, reflecting their unique approach to the problem and their understanding of Java programming.

But your task doesn't end with the generation of code. It's now time for the students to delve deeper, and here comes the importance of tracing in programming. Code tracing is the process of manually going through the execution of code, step-by-step, to understand its flow and logic. It's a fundamental practice in learning programming, aiding students in debugging their programs and developing a stronger understanding of code behavior.

One of the best ways to encourage code tracing is by generating tracing questions. Such questions stimulate the students to analyze their code at a granular level, fostering their problem-solving skills and enhancing their understanding of the Java language.

Remember, your questions should stem from two sources - the initial Requirement and the Code the students have produced. For example, given a code snippet, you might ask 'What is the output of this code if the user inputs two and three and chooses to multiply?' or 'Can this code handle the division by zero scenario, according to the requirement?'.

Your goal is not just to evaluate their coding skills, but to instill a deep, lasting understanding of Java programming in them. By asking relevant tracing questions, you're helping to nurture a new generation of critical thinkers, problem solvers, and proficient programmers.

You will ask many relevant tracing questions given 1. Requirement, 2. Code, as below:
Requirement: (blank) Code:

```java
int x = 3;
int y = 2;
if (x > 2)
   x++;
if (y > 1)
   y++;
if (x > 2)
   System.out.print("first ");
if (y < 3)
   System.out.print("second ");
System.out.print("third");
```

This is an example of our prompt.

## C  GPT3.5 vs GPT4

**Prompt** ... Requirement: (blank) Code:

```
int x = 3;
int y = 2;
if (x > 2)
   x++;
if (y > 1)
   y++;
if (x > 2)
   System.out.print("first ");
if (y < 3)
   System.out.print("second ");
System.out.print("third");
```

**GPT-3.5 Result:**

Tracing Questions:

1. What are the initial values of x and y in this code?

2. Does the if statement on line 3 execute? Why or why not?

3. Does the if statement on line 5 execute? Why or why not?

4. What is the value of x after line 4 has executed?

5. What is the value of y after line 6 has executed?

6. Which System.out.print statement(s) will execute in this code?

7. What is the output of this code?

8. What happens if the initial value of x is 1? How does the output change?

9. What happens if the initial value of y is 3? How does the output change?

10. Can this code handle other arithmetic operations besides addition and subtraction, according to the requirement?

**GPT-4 Result:** 1. What will be the values of x and y after executing the first two if statements?

2. What is the output of the code snippet as it is?

3. If we change the initial value of x to 1, what will be the output of the code snippet?

4. If we change the initial value of y to 0, what will be the output of the code snippet?

5. What will happen if both x and y are initialized with values less than or equal to 2? What will be the output in that case?

6. How would you modify the code to include an additional condition where it checks if x is equal to y and prints "equal" if it's true?

7. If we swap the positions of the second and third if statements, would the output change? Why or why not? 8. Can you explain the flow of the code execution given the initial values of x and y? What are the key decision points in the code?

9. What happens if we change the second if statement to "if (y == 1)"? How does that change the output of the code snippet?

10. How would you modify the code to print "none" if none of the conditions are met in the if statements?

This is an example of GPT3.5-turbo v.s. GPT4 for tracing question generation on a simple java problem.

**Prompt** ... Requirement: (blank) Code:

```
Requirement: (blank)
Code:
public static int redo(int i, int j)
{
    if (i==0)
        return 0;
    else
        return redo(i/j, j)+1;
}
```

**GPT-3.5 Result:**
1. What is the purpose of the redo() method?
2. What is the return type of the redo() method?
3. What is the initial value of i and j when the redo() method is called?
4. What happens if the value of i is 0?
5. What happens if the value of j is 0?
6. What happens if the value of i is negative?
7. What happens if the value of j is negative?
8. What happens if the value of i is greater than j?
9. What is the output of the redo() method if the values of i and j are 8 and 2 respectively?
10. What is the output of the redo() method if the values of i and j are 10 and 3 respectively?

**GPT-4 Result:**
1. What is the base case for this recursive function, and what does it return?
2. How is the recursion progressing towards the base case in this code?
3. What will be the output of this function if the inputs are i=16 and j=2?
4. What will be the output of this function if the inputs are i=81 and j=3?
5. How will the function behave if both i and j are equal to 0? Will it cause any issues?
6. How will the function behave if i is less than j? Is the output as expected according to the requirement?
7. How does the function handle the scenario when j is 1? Is the output as expected according to the requirement?
8. Can this function handle negative inputs for i and j? If not, how can it be modified to handle such cases?
9. What is the maximum recursion depth for this function, and how does it depend on the input values i and j?
10. Are there any edge cases or scenarios that this function might not handle well? If so, can you suggest any modifications to improve the code?

This is an example of GPT3.5-turbo v.s. GPT4 for tracing question generation on a recursion java problem.

## D   Zero-Shot vs Few-Shot in GPT4

**Zero-Shot Prompt** ... Requirement: (blank) Code:

```
Requirement: (blank)
Code:
public static int redo(int i, int j)
{
   if (i==0)
      return 0;
   else
      return redo(i/j, j)+1;
}
```

1.

```
Requirement: (blank)
Code:
for (int i = 3; i <= 9; i++)
{
   System.out.print(""*"");
}
Tracing question:
How many times does the following method print a *?
```

2.

```
Requirement: (blank)
Code:
//precondition: x >=0
public static void mystery (int x)
{
   System.out.print(x % 10);

   if ((x / 10) != 0)
   {
     mystery(x / 10);
   }
   System.out.print(x % 10);
}
Tracing question:
Given the following method declaration, which of the following is
   ↪   printed as the result of the call mystery(1234)?
```

3.

```
Requirement: (blank)
Code:
public static int product(int n)
{
   if (n <= 1)
      return 1;
   else
      return n * product(n - 2);
}
Tracing question:
Given the following method declaration, what value is returned as
   ↪  the result of the call product(5)?
```

Your goal is not just to evaluate their coding skills, but to instill a deep, lasting understanding of Java programming in them. By asking relevant tracing questions, you're helping to nurture a new generation of critical thinkers, problem solvers, and proficient programmers. You will ask many relevant tracing questions given 1. Requirement, 2. Code, as below. You don't have to strictly follow the examples, and we encourage you to be creative when thinking about the tracing questions: Requirement: (blank) Code:

```
Requirement: (blank)
Code:
public static int redo(int i, int j)
{
   if (i==0)
      return 0;
   else
      return redo(i/j, j)+1;
}
```

**Zero-Shot Result:**
1. What is the purpose of this function? What does it aim to accomplish?
2. What is the base case for this recursive function? What happens when the base case is reached?
3. If the input values are i = 8 and j = 2, what will be the output of the function? Please trace through the code step-by-step and explain your reasoning.
4. If the input values are i = 27 and j = 3, what will be the output of the function? Please trace through the code step-by-step and explain your reasoning.
5. What happens if the input values are i = 0 and j = 5? Does the function handle this case correctly? Explain your reasoning.
6. What happens if the input values are i = 5 and j = 0? Does the function handle this case correctly according to the requirement? Explain your reasoning.
7. What happens if both input values are negative, such as i = -8 and j = -2? Does the function handle this case correctly according to the requirement? Explain your reasoning.
8. Are there any edge cases or potential issues with the current implementation of the function? If so, what are they and how can they be addressed?

**Few-Shot Result:**
Tracing question 1: What is the output of the method redo(9, 3)?
Tracing question 2: What is the output of the method redo(0, 5)?
Tracing question 3: What is the output of the method redo(27, 3)?
Tracing question 4: How many times does the method redo call itself when i = 16 and j = 2?

Tracing question 5: Can this method handle cases where either i or j is a negative number? If so, what would be the output for redo(-6, 2)?

This is an example GPT4 in Few-Shot setting v.s. Zero-Shot Setting.