# OpenReview forum: "Exploring the Potential of Large Language Models in Generating Code-Tracing Questions for Introductory Programming Courses"
_EMNLP/2023/Conference — EMNLP 2023 Findings_

### Official Review · Reviewer_Qaqj · 2023-07-28

**Soundness:** 4

**Excitement:**

4: Strong: This paper deepens the understanding of some phenomenon or lowers the barriers to an existing research direction.

**Missing References:**

No important references appear to be missing.

**Paper Topic And Main Contributions:**

This paper explores the potential of using large language models (LLMs) to automatically generate code-tracing questions for introductory programming courses. Here are the key points:

- Problem addressed: Manually creating good code-tracing questions is time-consuming and lacks scalability. Existing automated methods using program analysis are limited. This paper investigates whether LLMs can generate high-quality and diverse code-tracing questions.

- Approach: The authors prompt GPT-4 to generate tracing questions based on code snippets and descriptions. They curate a dataset of human-authored questions for comparison.

- Metrics: A set of human evaluation metrics are introduced to assess quality of LLM questions versus human questions in terms of relevance, clarity, difficulty, etc. Textual similarity metrics are also used.

- Findings: Analysis shows GPT-4 can produce questions comparable in quality to human-authored ones. Over 50% of LLM questions were mistaken for human questions by experts. GPT-4 questions were sufficiently distinct from human questions.

- Contributions:

1) A high-quality dataset of human and LLM tracing questions.
2) Methodology and analysis to evaluate LLM's ability for this task.
3) Insights into the potential of LLMs for educational content generation.
4) A foundation for considering LLMs' inclusion in learning platforms.

In summary, this paper explores and evaluates the promise of LLMs for automatic high-quality code-tracing question generation, an important problem in computer science education. The human evaluation framework and dataset are valuable contributions.

**Questions For The Authors:**

Here are some questions I would have for the authors:

- You tested GPT-4 in a zero-shot setting without any demonstration questions. How would an few-shot approach comparing a wider range of prompting strategies affect the quality and diversity of generated questions?

- The human evaluation relied on subjective quality ratings. Were any objective measures like correctness considered? How can you further validate the appropriateness of questions?

- Have you considered testing this approach on other programming languages besides Java or other question types besides tracing? How robust is the approach to new domains?

- Were there any noticeable differences in the types of errors or limitations seen in GPT-4 questions compared to human questions?


- Do you have any thoughts on potential risks like teaching incorrect programming concepts if generated questions contain bugs? How can this be addressed?

**Reasons To Accept:**

This paper has several strengths that would make it a valuable contribution to the NLP community:

- Novel application area: This is the first work exploring LLMs for code tracing question generation, an important educational application distinct from prior NLP work. Expanding LLMs to this new domain is an interesting direction.

- Rigorous human evaluation: The paper introduces a thoughtful evaluation methodology including multiple human ratings and textual similarity metrics. This provides a model for human evaluation of LLM-generated content.

- High-quality dataset: The human question dataset compiled is a unique asset that can enable further research. Releasing this data would be very beneficial.

- Insightful analysis: The paper provides a nuanced analysis around the promise but also limitations of LLMs for this application. The insights into quality, diversity, and discernibility of LLM questions are impactful.

- Interdisciplinary relevance: This work sits at the intersection of NLP, education, and human-AI collaboration. Demonstrating the potential of LLMs for enhancing education would be appealing to a broad audience.

- Practical implications: With further development, the proposed techniques could lead to LLM integration in educational platforms and classrooms. This could significantly aid programming education and learning.

Presenting this work would expand the NLP community's understanding of LLMs' capabilities on a non-traditional task. It would also highlight an interesting new application domain with practical relevance. The novel human evaluation framework and high-quality dataset are additional assets that could catalyze further work in this direction. Overall, this is a timely exploration that would be well-suited for an NLP venue.

**Reasons To Reject:**

Some potential weaknesses and risks to consider:

- Small dataset size: The dataset has only 176 question pairs from limited sources. A larger, more diverse dataset could strengthen the evaluation and generalizability of findings.

- Limited scope: The focus is narrow - only on intro Java programming tracing questions. Testing breadth by targeting other domains/question types would be informative.

**Reproducibility:**

4: Could mostly reproduce the results, but there may be some variation because of sample variance or minor variations in their interpretation of the protocol or method.

**Reviewer Confidence:**

5: Positive that my evaluation is correct. I read the paper very carefully and I am very familiar with related work.

**Typos Grammar Style And Presentation Improvements:**

Overall the paper is very well written. I did not notice any major grammatical or typographical errors in the paper.

---

> ### Author Rebuttal · Authors · 2023-08-29
>
> We appreciate your review and feedback on our manuscript! Below we respond to your helpful questions and suggestions.
>
> ### Response to weaknesses:
> - Small data size: We appreciate you raising the important point about the current dataset size. As an initial study, our focus was an analysis of introductory Java tracing questions, especially those in a publicly available curriculum (CSAwesome). While expanding the dataset breadth and size is critical for future work this staged approach allows us to evaluate the methodology before larger classroom integration. Moreover, the evaluation expert screening process, requiring at least 1 year of programming teaching experience, helped mitigate the smaller dataset size. All 4 experts were computer science graduate students who received compensation for their thorough evaluation. Based on the promising results so far, we plan to apply our methodology to larger datasets with more diverse question types. This staged approach will allow us to evaluate the approach and ultimately integrate it into classrooms.
>
> - Limited scope: We intentionally focused on introductory Java tracing questions as a controlled initial domain to study. This allowed exploration of the nuances and challenges specific to code tracing. Now that we have established the methodology and initial findings, expanding to other programming languages and question types is a logical next step. Much of the human annotation protocol, training approach, and evaluation framework should be generalized to new domains with small adaptations. We agree broadening the scope will be important for future work to understand generalizability across different types of educational questions.
>
> ### Response to questions:
> (1) Few-Shot Prompting Approach: We appreciate the reviewer's suggestion to further explore few-shot prompting strategies. In preliminary experiments, we found that providing example questions introduced a bias that limited diversity compared to zero-shot prompting (see examples in Appendix D). Our initial human evaluation of the few-shot questions showed substantially lower diversity compared to zero-shot, with performance comparable to template-based methods.
>
> (2) Correctness metrics: In our study, we incorporated a metric where our experts evaluated whether each generated item qualified as a 'tracing question.' This was an important step in confirming the legitimacy of the generated questions in this particular category. While the overall quality of these generated tracing questions was found to be high, we opted not to overburden our expert evaluators with multiple criteria for correctness. Instead, we introduced a 'discernibility' metric, aiming to investigate whether the questions could be distinguished as either human-authored or generated by an LLM. We believe that the results of this discernibility metric could offer insights into the quality and applicability of the generated questions. The evaluation form also contains quality ratings by experts on relevance, clarity, difficulty, etc. using a standardized rubric.
>
> In terms of established automatic metrics, our methodology employed well-regarded measures such as BLEU (Post, 2018), ROUGE (Lin, 2004), and rubrics from (Kurdi et al., 2020). These measures aimed to assess key attributes like question quality, diversity, and relevance. We also utilized textual similarity metrics for automatic evaluation to gauge the diversity of generated questions. If you have any specific recommendations on such metrics, we would be deeply appreciative of your expert advice.
>
> (3) New Domains: Testing the approach on other programming languages and question types is an excellent suggestion. We focused on intro Java tracing questions as an initial study, but evaluating robustness to new domains will be important in future work. We can reuse much of the methodology and infrastructure from this paper when expanding the scope.
>
> (4) Differences in questions: We acknowledge that human-authored and LLM-generated questions exhibit distinct characteristics. As discussed in our discernibility study (Section 4.2, Table 3), it was very difficult for expert annotators to discern whether individual questions were authored by humans or GPT-4. However, when analyzing question quality holistically (Table 1), human-authored questions were more likely to receive high scores. This suggests nuanced differences exist between the question types. To deeply assess these differences, we have initiated a separate study focused on comparing the types and quality of LLM-generated and human-authored questions. While this analysis is still underway, preliminary results reveal insights into the limitations and biases of LLM question generation. Your question underscores the importance of this direction, and we aim to provide a detailed characterization of the differences in a follow-up work.
>
> (5) Risks: You raise a critical point about potential risks like propagating incorrect concepts if LLM questions contain bugs. Rigorous vetting by experts is an important safeguard we should adopt before any real-world deployment. A system that deploys such questions in a classroom environment should also include checks using test suites and static analysis to catch bugs prior to student exposure. Minimizing the risk of students learning incorrect information is an ethical priority we agree is necessary to explore.
>
> We appreciate these insightful suggestions for strengthening the work. We will work towards incorporating this valuable feedback to improve our methodology and analysis in future extensions. Please let us know if we can address any other questions!

---

### Official Review · Reviewer_TK2P · 2023-07-30

**Soundness:** 3

**Excitement:**

2: Mediocre: This paper makes marginal contributions (vs non-contemporaneous work), so I would rather not see it in the conference.

**Paper Topic And Main Contributions:**

This paper investigates the application of LLMs for outputing code-tracing questions in indroductory prgramming courses.

**Questions For The Authors:**

See reasons to reject

**Reasons To Accept:**

- The code-tracing scenario proposed by this paper is novel.
- A dataset has been constructed for this scenario.
- An exploration of LLM's application in this scenario.
- The metrics has been established to evaluate the quality of the generated code-tracing questions.

**Reasons To Reject:**

- The study only considered GPT-4 and has not incorporated other models.
- There is no personalized question recommendation based on different people's submissions, which shows a lack of thorough exploration of the scenario.
- The study lacks attempts to use this approach in practical testing scenarios.
- The metrics for evaluating code-tracing questions are not comprehensive enough.
- The setting of the metrics is somewhat subjective and lacks an automatic evaluation.

**Reproducibility:**

4: Could mostly reproduce the results, but there may be some variation because of sample variance or minor variations in their interpretation of the protocol or method.

**Reviewer Confidence:**

5: Positive that my evaluation is correct. I read the paper very carefully and I am very familiar with related work.

---

> ### Author Rebuttal · Authors · 2023-08-29
>
> We appreciate your review and feedback on our manuscript! We would like to address some of the concerns raised and provide further clarifications:
>
> 1. Our choice of GPT-4 was based on it being the current state-of-the-art model for code generation and understanding, as evidenced by leaderboards like HumanEval (https://paperswithcode.com/sota/code-generation-on-humaneval). Our initial experiments found GPT-4 generated more diverse questions than GPT-3.5 turbo for the same prompts, as measured by BERTScore (Section 3.3, Figure 2). Examples in Appendix D illustrate GPT-4's greater range of tracing questions. While we only evaluated GPT-4, future work could incorporate comparisons to other recently released models. We chose GPT-4 because of its superior performance on code understanding and also because it does not require local computing resources, making it easily accessible for education practitioners, or researchers who would like to explore this topic.
>
> 2. We appreciate your suggestion regarding the personalization of tracing questions based on student submissions. In this initial study, our focus was to compare LLM-generated questions with expert-authored ones from established curricula like CSAwesome. This provides a crucial baseline for quality and methodology that we believe sets the stage for future personalization work. Insights from this foundational study will inform upcoming efforts to tailor questions to individual student code submissions, as well as practical studies involving learner engagement. Thank you for highlighting this important avenue for research.
>
> 3. We acknowledge the importance of practical testing scenarios, such as classroom experiments for this type of educational work. Our initial analysis focused on the assessment of the questions’ quality as we believe this is a necessary step before practical testing can be implemented by deploying LLM-generated questions in the classroom. For future work, our ongoing project focuses on the design and construction of an education platform implemented with tracing questions generated by LLMs, usability testing with students, and an evaluation of learning outcomes. We consider a real classroom setting as the immediate step 2 after this study.
>
> 4. (and 5.) To comprehensively evaluate the quality of generated questions, we adopt both automatic evaluation with a variety of metrics and human evaluation with objective and subjective ratings. Evaluation metrics used in this study:
> * Automatic/Objective Evaluation: 1) BERTScore similarity metric comparing GPT-4 and GPT-3.5 questions for the same prompts. This showed GPT-4 had higher diversity. 2) Textual similarity metrics like BLEU, ROUGE, and BERTscore compared GPT-4 questions to human questions.
>  * Subjective Evaluation: 1) Expert preferences for best questions when comparing GPT-4 and human-authored pairs. 2) Discernibility of authorship using confusion matrix of expert guesses. 3) Quality ratings by experts on relevance, clarity, difficulty, etc. using a standardized rubric.
>
> We selected our evaluation metrics based on established literature, including BLEU (Post, 2018) for automatic evaluation. Our automatic evaluations assessed textual similarity and diversity through metrics like BLEU, ROUGE, and BERTScore. The human evaluation criteria roughly followed (Kurdi et al., 2020) to cover multiple aspects like relevance, clarity, difficulty appropriateness, and quality. For the purpose of our study, we further split relevance into 1) relevance to code snippets, and 2) relevance to learning objectives. In the perspective of quality, “is this a tracing question” was used to evaluate the validity of a tracing question. These were rated on a standardized 1-5 rubric by 4 experts, who were screened with criteria such as being computer science graduate students with at least 1 year of programming teaching/tutoring experience.
>
> In summary, our work establishes the preliminary steps in a novel and underexplored task. While we recognize areas of potential extension, we believe that the findings presented in this manuscript are an important first step in demonstrating LLMs' capacity for automated code-tracing question generation. These insights are not only valuable for researchers but are immediately implementable by educational practitioners in their instructional projects.
>
>
> ### References:
>
> [1] Post, M. (2018). A Call for Clarity in Reporting BLEU Scores. WMT 2018, 186.
>
> [2] Kurdi, G., Leo, J., Parsia, B., Sattler, U., & Al-Emari, S. (2020). A systematic review of automatic question generation for educational purposes. International Journal of Artificial Intelligence in Education, 30, 121-204.

---

### Official Review · Reviewer_t9s4 · 2023-08-05

**Typos Grammar Style And Presentation Improvements:** Table 1 and Table 2 can be merged.
**Soundness:** 2

**Excitement:**

2: Mediocre: This paper makes marginal contributions (vs non-contemporaneous work), so I would rather not see it in the conference.

**Paper Topic And Main Contributions:**

The study uses large language models (LLMs) to generate code-tracing questions for learning a programming language. Results demonstrate that GPT4 has an acceptable ability and potential to generate questions for code-tracing and help programming education. The generated data can be one of the contributions to the field of programming education and developing a dialogue tutoring system.

**Questions For The Authors:**

A. Please elaborate more details in Section 4.3, that is, does BLEU refer to BLEU4? What does BERTScore mean?

**Reasons To Accept:**

* The work well demonstrates how to prompt LLMs for a series of code-tracing questions when given a snippet of codes.
* When the generated data is released, it may bring benefits to the field of programming education and dialogue systems.


**Reasons To Reject:**

* The experiment results show the potential and limitations of the LLMs-generated questions for code-tracing.


**Reproducibility:**

4: Could mostly reproduce the results, but there may be some variation because of sample variance or minor variations in their interpretation of the protocol or method.

**Reviewer Confidence:**

3: Pretty sure, but there's a chance I missed something. Although I have a good feel for this area in general, I did not carefully check the paper's details, e.g., the math, experimental design, or novelty.

---

> ### Author Rebuttal · Authors · 2023-08-29
>
> We appreciate the reviewer summarizing both the potential and limitations observed in our study. As a focused contribution, our work made the following key contributions:
>
> * The curation of a high-quality dataset consisting of education experts and LLM-generated code tracing questions and associated code snippets.
> * An exploration and evaluation of GPT4’s capability in question generation, including comparisons with both GPT3.5-turbo and human-authored questions, and an examination of few-shot and zero-shot scenarios.
> * The introduction of a human evaluation methodology and a comprehensive assessment of the quality of LLM-generated questions, offer valuable insights into the potential of LLMs in educational contexts.
>
> Through multi-faceted evaluations, our results demonstrated that LLMs can produce tracing questions comparable to human-authored ones in quality, while exhibiting greater diversity. The low discernibility highlights LLMs' ability to generate questions that experts could not reliably distinguish from human-authored questions. This provides evidence that LLMs can create tracing questions independently rather than just paraphrasing human questions. We believe these findings meaningfully demonstrate the potential of LLMs for this underexplored application.
>
> We also acknowledge the study's limitations, including the focus on a specific dataset, the lack of personalized recommendations, and the absence of classroom evaluations. As you rightly noted, these aspects represent interesting future work that can build on the foundations we have established here. As a short paper, our scope was a focused analysis that acts as a first important step towards incrementally addressing these limitations in follow-up studies.
>
> Please do not hesitate to suggest any other future directions that you think could be worthwhile.
>
> Questions Answered:
>
> Q1: In Section 4.3, Does BLEU refer to BLEU4?
>
> Answer: Yes, the BLEU metric used in Section 4.3 refers specifically to BLEU-4. This is the 4-gram BLEU score, which calculates n-gram precision for up to 4-grams. We use nltk sentence_bleu for implementation. We would be happy to add these clarifications to our manuscripts
>
> Q2: What does BERTScore mean?
>
> Answer: We appreciate these questions and would be happy to add additional information to our manuscript to clarify the meaning and the use of BERTScore. BERTScore measures the similarity between two texts based on their BERT embeddings. It uses a pre-trained BERT model to extract contextual embeddings for each input text, and then calculates similarity using precision, recall, and F1.
>
> Specifically in the paper, BERTScore was used in two ways:
> 1. Section 3.3. To compare the diversity of GPT-3.5 and GPT-4 - In Figure 2, We draw a box plot of BERTScores for the many generated questions with regard to one reference. GPT-4's box plot showed a wider distribution of BERTScores. This indicates GPT-4 produced questions with more lexical diversity compared to GPT-3.5, reflecting GPT-4's ability to generate more diverse questions.
> 2. Section 4.3. BERTScore was used to compare the semantic similarity between the GPT-4 generated questions being evaluated in this study and the reference human-authored questions. In Table 4, The moderate BERTScore F1 of 0.303 suggests the GPT-4 questions capture similar semantics but with different text than the human questions.

---

### Meta-Review · Area_Chair_qUM5 · 2023-09-15

**Recommendation:** 4

**Metareview:**

The reviewers find that this paper offers valuable contributions by exploring the potential of LLMs in code-tracing question generation for programming education, backed by a robust methodology and insightful analysis. It addresses a novel application area with interdisciplinary relevance, making it a strong candidate for acceptance in an NLP venue.

The authors raised some concerns with the review submitted by Reviewer 1, which does not seem to be of good quality (the scores are low, but the actual reasons to reject are missing). Moreover, the reviewers highlight reasons to reject such as lack of personalization and narrow scope (Java programming), which have more to do with the limitations of the paper (potential future work) than with the soundness of the described methodology.

Paper Topic And Main Contributions:

This paper investigates the potential of using large language models (LLMs), specifically GPT-4, to automatically generate code-tracing questions for introductory programming courses. The study addresses the challenge of time-consuming and limited scalability in manually creating such questions and explores whether LLMs can provide a solution. The authors employ GPT-4 to generate questions based on code snippets and descriptions, comparing them to human-authored questions using various evaluation metrics. Notably, the results reveal that GPT-4 can produce questions of comparable quality to those crafted by humans, with over 50% being indistinguishable from human-authored questions. The contributions of this research include a high-quality dataset of both human and LLM-generated code-tracing questions, a rigorous evaluation methodology, insights into the potential of LLMs for educational content generation, and a foundation for considering their integration into learning platforms. In essence, this paper offers a promising solution to the challenge of automating code-tracing question generation in computer science education, providing a valuable dataset and evaluation framework in the process.

Reasons To Accept:

* Novelty and Benefits for Educational Applications
    * The paper introduces a novel application of LLMs in code-tracing question generation, offering a unique perspective distinct from previous NLP work.
    * The generated data, when released, has the potential to bring significant benefits to the field of programming education and dialogue systems, addressing an important educational need.
* Quality of the Methodology
    * The authors have constructed a high-quality dataset tailored to the code-tracing scenario, which can serve as a valuable resource for future research.
    * The introduction of a rigorous evaluation methodology, including human ratings and textual similarity metrics, sets a standard for assessing LLM-generated content, contributing to the field's evaluation practices.
* Quality of the Analyses
    * The paper provides insightful analysis, delving into the promise and limitations of LLMs in generating code-tracing questions, offering valuable insights into quality, diversity, and discernibility.

Reasons To Reject:

* The study solely focuses on GPT-4 for code-tracing question generation and does not incorporate other LLMs, limiting the exploration of potential alternatives and comparative analysis.
* The dataset used in the study consists of only 176 question pairs from limited sources, which raises concerns about the statistical robustness and generalizability of the findings. A larger and more diverse dataset is needed for a more reliable evaluation.
* The metrics used for evaluating code-tracing questions are not comprehensive enough, and the setting of these metrics is somewhat subjective, lacking an automatic evaluation component.

---

### Decision · Program_Chairs · 2023-10-07

**Decision:**

Accept-Findings

**Comment:**

The reviewers find that this paper offers valuable contributions by exploring the potential of LLMs in code-tracing question generation for programming education, backed by a robust methodology and insightful analysis. It addresses a novel application area with interdisciplinary relevance, making it a strong candidate for acceptance in an NLP venue.

The authors raised some concerns with the review submitted by Reviewer 1, which does not seem to be of good quality (the scores are low, but the actual reasons to reject are missing). Moreover, the reviewers highlight reasons to reject such as lack of personalization and narrow scope (Java programming), which have more to do with the limitations of the paper (potential future work) than with the soundness of the described methodology.

Paper Topic And Main Contributions:

This paper investigates the potential of using large language models (LLMs), specifically GPT-4, to automatically generate code-tracing questions for introductory programming courses. The study addresses the challenge of time-consuming and limited scalability in manually creating such questions and explores whether LLMs can provide a solution. The authors employ GPT-4 to generate questions based on code snippets and descriptions, comparing them to human-authored questions using various evaluation metrics. Notably, the results reveal that GPT-4 can produce questions of comparable quality to those crafted by humans, with over 50% being indistinguishable from human-authored questions. The contributions of this research include a high-quality dataset of both human and LLM-generated code-tracing questions, a rigorous evaluation methodology, insights into the potential of LLMs for educational content generation, and a foundation for considering their integration into learning platforms. In essence, this paper offers a promising solution to the challenge of automating code-tracing question generation in computer science education, providing a valuable dataset and evaluation framework in the process.

Reasons To Accept:

* Novelty and Benefits for Educational Applications
    * The paper introduces a novel application of LLMs in code-tracing question generation, offering a unique perspective distinct from previous NLP work.
    * The generated data, when released, has the potential to bring significant benefits to the field of programming education and dialogue systems, addressing an important educational need.
* Quality of the Methodology
    * The authors have constructed a high-quality dataset tailored to the code-tracing scenario, which can serve as a valuable resource for future research.
    * The introduction of a rigorous evaluation methodology, including human ratings and textual similarity metrics, sets a standard for assessing LLM-generated content, contributing to the field's evaluation practices.
* Quality of the Analyses
    * The paper provides insightful analysis, delving into the promise and limitations of LLMs in generating code-tracing questions, offering valuable insights into quality, diversity, and discernibility.

Reasons To Reject:

* The study solely focuses on GPT-4 for code-tracing question generation and does not incorporate other LLMs, limiting the exploration of potential alternatives and comparative analysis.
* The dataset used in the study consists of only 176 question pairs from limited sources, which raises concerns about the statistical robustness and generalizability of the findings. A larger and more diverse dataset is needed for a more reliable evaluation.
* The metrics used for evaluating code-tracing questions are not comprehensive enough, and the setting of these metrics is somewhat subjective, lacking an automatic evaluation component.